# Learning Mutational Semantics

**Brian Hie**
MIT
brianhie@mit.edu

**Ellen D. Zhong**
MIT
zhonge@mit.edu

**Bryan D. Bryson**
MIT
bryand@mit.edu

**Bonnie Berger**
MIT
bab@mit.edu

## Abstract

In many natural domains, changing a small part of an entity can transform its semantics; for example, a single word change can alter the meaning of a sentence, or a single amino acid change can mutate a viral protein to escape antiviral treatment or immunity. Although identifying such mutations can be desirable (for example, therapeutic design that anticipates avenues of viral escape), the rules governing semantic change are often hard to quantify. Here, we introduce the problem of identifying mutations with a large effect on semantics, but where valid mutations are under complex constraints (for example, English grammar or biological viability), which we refer to as constrained semantic change search (CSCS). We propose an unsupervised solution based on language models that simultaneously learn continuous latent representations. We report good empirical performance on CSCS of single-word mutations to news headlines, map a continuous semantic space of viral variation, and, notably, show unprecedented zero-shot prediction of single-residue escape mutations to key influenza and HIV proteins, suggesting a productive link between modeling natural language and pathogenic evolution.[1]

## 1 Introduction

Much of the effort devoted to learning machine-intelligible representations of natural language semantics has been built on the "distributional hypothesis," in which the context and co-occurrence of words is assumed to provide insight into the meaning of words [25, 22, 35, 38, 41, 43]. While distributional semantics was developed to model human intuitive notions of "meaning," similar reasoning may be useful for domains beyond human intuition.

For example, like linguistic semantics, biological function is encoded by a sequence of tokens (the bases of nucleic acids or the amino acid residues of proteins) that is determined by a complex distributional structure. Promisingly, recent analyses of biological sequence inspired by tools for modeling natural language have been shown to improve prediction of biological function [9, 45, 5].

A pressing and still poorly understood biological problem is understanding how rapidly mutating viral proteins can evade recognition by "escaping" the immune system's antibodies. Viral escape, which can be caused by even a single-residue change, has prevented the development of a universal antibody-based vaccine for influenza [30, 33] or human immunodeficiency virus (HIV) [6]. However, the rules governing viral fitness are complex and a biological experiment that empirically tests the escape potential of all mutations to all viral strains would be prohibitively expensive. A key concept underlying this study is that, in order to escape the immune system, a mutation must not only preserve

viral infectivity (i.e., it must be "grammatical") but it must also be functionally altered so that it is no longer recognized by the immune system's antibodies (i.e., it must have substantial "semantic change").

Here, we introduce the problem of searching for sequence mutations based on both high semantic change and grammatical validity, which we call constrained semantic change search (CSCS). This is in contrast to settings concerned with semantic similarity search, rather than change. To gain intuition, we apply CSCS to natural language and, to demonstrate broader impact, we apply CSCS to predict viral escape. Our key contributions are **(1)** we introduce the CSCS problem formulation and show how learned language models offer a compelling solution with strong empirical results on both natural language and biological applications, suggesting that the distributional hypothesis from linguistics is also useful for modeling pathogenic evolution; **(2)** we develop an unsupervised neural language model for viral proteins and show that it learns semantically meaningful embeddings; and **(3)** we use CSCS for zero-shot prediction of escape mutations for influenza and for HIV with quantitative results much higher than baseline methods. To our knowledge, we present the first computational model that effectively predicts viral escape, potentially enabling vaccine or therapeutic design that anticipates escape before it occurs.

## 2 Methods

### 2.1 Problem Formulation

Intuitively, our goal is to identify mutations that induce high semantic change (e.g., a large impact on biological function) while being grammatically acceptable (e.g, biologically viable). More precisely, we are given a sequence of tokens defined as $\mathbf{x} \triangleq (x_1, ..., x_N)$ such that $x_i \in \mathcal{X}, i \in [N]$, where $\mathcal{X}$ is a finite alphabet (e.g., characters or words for natural language, or amino acids for protein sequence). Let $\tilde{x}_i$ denote a mutation at position $i$ and the mutated sequence as $\mathbf{x}[\tilde{x}_i] \triangleq (..., x_{i-1}, \tilde{x}_i, x_{i+1}, ...)$.

We first require a semantic embedding $\mathbf{z} \triangleq f_s(\mathbf{x})$, where $f_s : \mathcal{X}^N \to \mathbb{R}^K$ embeds discrete-alphabet sequences into a continuous space, where, ideally, closeness in embedding space would correspond to semantic similarity. We denote semantic change as the distance in embedding space, i.e.,

$$\Delta \mathbf{z}[\tilde{x}_i] \triangleq \|\mathbf{z} - \mathbf{z}[\tilde{x}_i]\| = \|f_s(\mathbf{x}) - f_s(\mathbf{x}[\tilde{x}_i])\| \qquad (1)$$

where $\|\cdot\|$ denotes a vector norm. The grammaticality of a mutation is described by

$$p(\tilde{x}_i|\mathbf{x}), \qquad (2)$$

which takes values close to zero if $\mathbf{x}[\tilde{x}_i]$ is not grammatical and close to one if it is grammatical.

Our objective combines semantic change and grammaticality as a linear combination

$$a(\tilde{x}_i; \mathbf{x}) \triangleq \Delta \mathbf{z}[\tilde{x}_i] + \beta p(\tilde{x}_i|\mathbf{x})$$

for each possible mutation $\tilde{x}_i$ and a user-specified parameter $\beta \in [0, \infty)$. Mutations $\tilde{x}_i$ are prioritized based on $a(\tilde{x}_i; \mathbf{x})$. We refer to ranking mutations based on semantic change and grammaticality as CSCS.

### 2.2 Algorithms

#### 2.2.1 Language Modeling

Algorithms for CSCS could potentially take many forms; for example, separate algorithms could be used to compute $\Delta \mathbf{z}[\tilde{x}_i]$ and $p(\tilde{x}_i|\mathbf{x})$ independently, or a two-step approach might be possible that computes one of the terms based on the value of the other.

Instead, we reasoned that a single approach could compute both terms simultaneously, based on learned language models that learn the probability distribution of a word given its context [38, 15, 43, 16, 44]. The language model we use throughout our experiments considers the full sequence context of a word and learns a latent variable probability distribution $\hat{p}$ and function $\hat{f}_s$, where, for all $i \in [N]$,

$$\hat{p}(x_i|\mathbf{x}_{[N]\setminus\{i\}}, \hat{\mathbf{z}}_i) = \hat{p}(x_i|\hat{\mathbf{z}}_i) \quad \text{and} \quad \hat{\mathbf{z}}_i = \hat{f}_s(\mathbf{x}_{[N]\setminus\{i\}}),$$

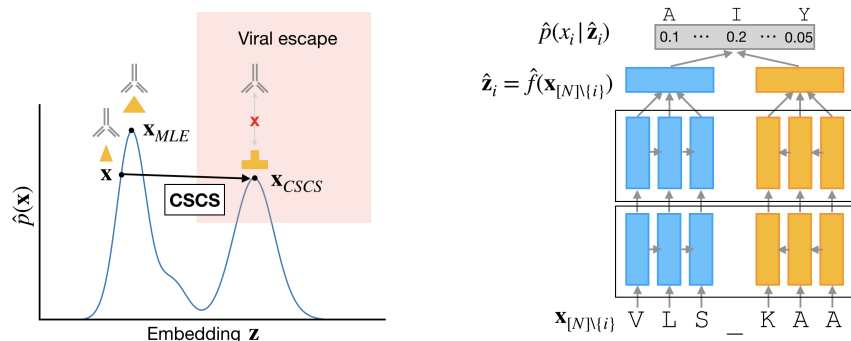

Figure 1: Constrained semantic change search (CSCS) for viral escape prediction. Left: Given an input sequence $\mathbf{x}$ and its semantics encoded by $\mathbf{z}$, CSCS aims to find a mutation to $\mathbf{x}_{\text{CSCS}}$ that causes the largest semantic change (high $\Delta\mathbf{z}$), while remaining grammatical (high $\hat{p}(\mathbf{x})$). Right: Language model architecture with two stacked BiLSTM layers instantiating the semantic embedding function $\hat{f}$, with the final language model output used as grammaticality.

i.e., latent variable $\hat{\mathbf{z}}_i$ encodes the context $\mathbf{x}_{[N]\setminus\{i\}} \triangleq (..., x_{i-1}, x_{i+1}, ...)$ such that $x_i$ is conditionally independent of its context given the value of $\hat{\mathbf{z}}_i$.

We use different aspects of the language model to describe semantic change and grammaticality by setting terms (1) and (2) as

$$\Delta\mathbf{z}[\tilde{x}_i] \triangleq \|\hat{\mathbf{z}} - \hat{\mathbf{z}}[\tilde{x}_i]\|_1 \quad \text{and} \quad p(\tilde{x}_i|\mathbf{x}) \triangleq \hat{p}(\tilde{x}_i|\hat{\mathbf{z}}_i),$$

where $\hat{\mathbf{z}} \triangleq \begin{bmatrix} \hat{\mathbf{z}}_1^{\mathrm{T}} & \cdots & \hat{\mathbf{z}}_N^{\mathrm{T}} \end{bmatrix}^{\mathrm{T}}$ is the concatenation of embeddings for each token, $\hat{\mathbf{z}}[\tilde{x}_i]$ is defined similarly but for the mutated sequence, and $\|\cdot\|_1$ is the $\ell_1$ norm, chosen because of more favorable properties compared to other standard distance metrics, though other metrics could be empirically quantified in future work [2].

Effectively, distances in embedding space are used to approximate semantic change and the emitted probability approximates grammaticality. We note that these modeling assumptions are not guaranteed to be perfectly specified, since, in the natural language setting for example, antonyms may also be close in embedding space and the language model output can also encode linguistic pragmatics in addition to grammaticality. However, we still find these modeling assumptions to have good empirical support.

Training or parameterizing the language model is separate from CSCS, and the novelty of CSCS is in leveraging these models in a new way. An advantage of this approach is that it does not require any bespoke modifications to the general language modeling framework, other than requiring a continuous latent variable. CSCS can therefore leverage the noted multitask generality of language models [44].

Importantly, this approach to CSCS is completely unsupervised. Rather than assume access to labels explicitly encoding semantics or grammaticality, the model instead extracts this information from a large unlabeled corpus. This is critical in domains, like viral genomics, in which large sequence corpuses are available but functional profiling is limited. These corpuses implicitly contain information related to grammaticality or infectivity (e.g., all sequences are grammatically acceptable or come from infectious virus), but the algorithm must learn these rules from data.

### 2.2.2 Architecture

Based on the success of recurrent architectures for protein-sequence representation learning [9, 45, 5], we use similar encoder models for viral protein sequences (**Figure 1**). Our model passes the full context sequence into bidirectional long-short-term-memory (BiLSTM) hidden layers. We used the concatenated output of the final LSTM layers as the semantic embedding, i.e.,

$$\hat{\mathbf{z}}_i \triangleq \begin{bmatrix} \text{LSTM}_f(g_f(x_1, ..., x_{i-1}))^{\mathrm{T}} & \text{LSTM}_r(g_r(x_{i+1}, ..., x_N))^{\mathrm{T}} \end{bmatrix}^{\mathrm{T}}$$

where $g_f$ is the output of the preceding forward-directed layer, $\text{LSTM}_f$ is the final forward-directed LSTM layer, and $g_r$ and $\text{LSTM}_r$ are the corresponding reverse-directed components. The final output

probability is a softmax-transformed linear transformation of $\hat{\mathbf{z}}_i$, i.e.,

$$\hat{p}(x_i|\mathbf{x}_{[N]\setminus\{i\}}) \triangleq \text{softmax}(\mathbf{W}\hat{\mathbf{z}}_i + \mathbf{b})$$

for some learned model parameters $\mathbf{W}$ and $\mathbf{b}$. In our experiments, we used a 20-dimensional dense embedding for each element in the alphabet $\mathcal{X}$, two BiLSTM layers with 512 units, and categorical cross entropy loss optimized by Adam with a learning rate of 0.001, $\beta_1 = 0.9$, and $\beta_2 = 0.999$. Additional details on hyperparameter selection are given in Appendix 6.3.1.

### 2.2.3 Rank-Based Acquisition

Rather than acquiring mutations based on raw semantic change and grammaticality values, which may be on very different scales, we find that selecting $\beta$ is much easier in practice when first rank-transforming the semantic change and grammaticality terms, i.e., acquiring based on

$$a'(\tilde{x}_i; \mathbf{x}) \triangleq \text{rank}(\Delta\mathbf{z}[\tilde{x}_i]) + \beta\,\text{rank}(p(\tilde{x}_i|\mathbf{x})).$$

All possible mutations $\tilde{x}_i$ are then given priority based on the corresponding values of $a'(\tilde{x}_i; \mathbf{x})$, from highest to lowest. Our empirical results have consistently good performance by simply setting $\beta = 1$ (equally weighting both terms), which we used in all experiments below unless otherwise noted. In this study, we deal with the unsupervised setting where $\beta$ is a parameter but note that adding some supervision could learn $\beta$ (or other, non-rank, transformations) from data.

### 2.2.4 Connection to Viral Escape

A language model is a probability distribution over sequences learned from a corpus of data. For any sequence $\mathbf{x}$, the model will output a predicted probability $p(\mathbf{x})$ of observing that sequence in the training data distribution. We call $p(\mathbf{x})$ "grammaticality" because in natural language tasks, $p(\mathbf{x})$ tends to be high for grammatically correct sentences. In the case of viral sequences, the training distribution consists of viral proteins that have evolved for high fitness/virality, so we hypothesize that high grammaticality corresponds to high viral fitness.

However, high fitness alone does not indicate an escape mutation. For example, a viral protein with a neutral mutation will have equally high fitness but may not look different enough to escape detection by the immune system, i.e., it will have no "antigenic" change. To identify mutations that do lead to large antigenic changes, we exploit the internal sequence embeddings learned by the language model. If two sequences have similar embeddings, then they have similar distributions over sequence continuations given the input tokens. As a natural-language example, "the men advance", "the soldiers advance", and "the three advance" have a similar set of possible word continuations and would have similar embeddings, while "the cash advance" has a nearly disjoint set of continuations and thus a different embedding. We hypothesize that neutral mutations should not affect the distribution over amino acids at other positions, while mutations that affect antigenicity do affect the distribution over other positions. Thus, the combination of high sequence probability (high fitness) and a large change in embedding (antigenic change) indicates an escape mutation.

## 3 Related Work

The CSCS problem is related to work focused on identifying the best interventions to structured data to produce a desired outcome [40, 42]. Such work often assumes a dataset that includes both the observed features and corresponding outcomes, which allows for supervised learning. In contrast, we assume no explicit labels of semantic change and must resort to unsupervised learning to extract this information. This is because in domains like viral mutation, data that directly measures viral fitness is very limited, while unlabeled sequence data is abundant.

Importantly, our CSCS task is distinct from representation learning tasks that construct semantically meaningful embeddings, but CSCS does stand to benefit from innovation in representation learning. Using hidden states in a language model to represent natural language semantics has been an influential and productive idea [43]. Rather than acquiring mutations based on greatest semantic change as in CSCS, acquisition based instead only on lowest $\Delta\mathbf{z}[\tilde{x}_i]$ essentially performs semantic similarity search among all sequences that differ by a single token.

In biological applications, neural language models have been developed to learn unsupervised or weakly supervised protein sequence embeddings that encode generic protein similarity [9, 45, 5]. To

```
Original:  australian dead in bali      Original:  winegrowers revel in good season
CSCS:      australian ballet in bali     CSCS:      winegrowers revel in flu season
              Original:  nauru bans transhipments to tackle overfishing
              CSCS:      nauru bans continue to tackle overfishing
```

Figure 2: Example CSCS-proposed mutations to news headlines show large changes to the headline meaning or to the syntactic part-of-speech structure.

our knowledge, however, no previous work has considered how mutations affect these embeddings, nor have such methods been applied to evolutionary change. Furthermore, while many variants of recurrent or transformer-based architectures have been proposed for protein sequence modelling tasks, we note any such current or future language model architecture could be used in CSCS.

Some work in computational biology has focused on identifying deleterious mutations in human or mammalian genomes with clinical relevance [51, 46]. However, these approaches are based on direct supervision under the assumption that rare or poorly conserved mutations are deleterious. Such an assumption, however, does not apply to escape mutations, which could be both frequent or infrequent in a population. Viral genomes are also more highly variable than mammalian genomes (e.g., "Drake's rule"), so aligning mutations across viral strains is more difficult [20, 14, 48].

Most computational analyses specific to viral mutation require rich metadata beyond raw sequence or make virus-specific assumptions [8, 54] (for example, vaccine-related temporal patterns in influenza, which are absent for HIV). Most similar to our approach, models exist for learning viral fitness from a large sequence corpus [27, 26]. These approaches, however, requires time-consuming and error-prone multiple sequence alignment (MSA) preprocessing [29] and only consider pairwise information couplings among residues, which, as demonstrated below, limit performance when predicting escape. To our knowledge, our work is the first to effectively model viral escape that generalizes to any relevant genomic sequence from diverse viruses, without the need for sequence alignment, complex metadata, or special assumptions on mutational processes.

## 4  Results

To demonstrate how CSCS can alter semantics while preserving grammaticality, we gain intuition by first applying CSCS in a natural language setting before demonstrating broader impact by applying CSCS to biological sequence mutation in viruses. We find that CSCS-mutated headlines are semantically altered (quantified via changes in part-of-speech (POS) structure and distance in WordNet hierarchy) while remaining grammatical. Using a language model trained on a large corpus of influenza sequences, we find that CSCS-mutated viral sequences are predictive of escape mutations (i.e., "grammatical" mutations that preserve biological viability and infectivity but that also alter the protein's "semantics" thereby enabling escape from vaccines or treatments) that were identified by independent biological experiments. To assess generality, we perform this zero-shot escape prediction in two different influenza subtypes and in HIV.

### 4.1  News Headlines

**Setup and Training Data.**  We sought to confirm our intuitions of "semantic change" and "grammaticality" by applying CSCS to single-word changes in news headlines. Our training corpus consisted of 1,186,018 headlines from the Australian Broadcasting Corporation from 2003 through 2019 (Appendix 6.1.1) [34].

**Language Model Selection.**  We selected our model architecture by holding out a test set of headlines from 2016 onward (179,887 headlines, about 15%) and evaluating cross entropy loss for the language modeling task. We used a cross-validation strategy within the training set to grid search hyperparameters (Appendix 6.3.1). Our BiLSTM model with access to the full context (described above) obtained a training and test loss of 2.2 and 6.0, respectively. Performance decreased when replacing the LSTM hidden layers with densely-connected layers (train loss = 2.3, test loss = 7.2) or when removing access to the right context, i.e., a language model task $p(x_i|\mathbf{x}_{[i-1]})$ (train loss = 4.2, test loss = 6.5).

Table 1: Headline Semantic Change Results.

| Setting | Median % POS Change | | Median WordNet Similarity | |
|---|---|---|---|---|
| | NLTK | FLAIR | Pathwise | Wu-Palmer |
| Semantically closest (smallest $\Delta\mathbf{z}[\tilde{x}_i]$) | 0.00% | 0.00% | 0.143 | 0.546 |
| CSCS-proposed (highest $a'(\tilde{x}_i; \mathbf{x})$) | 16.7% | 14.3% | 0.0833 | 0.235 |
| two-sided $t$-test $P$ | $<10^{-308}$ | $<10^{-308}$ | $<10^{-308}$ | $<10^{-308}$ |

Table 2: Grammatical Acceptability Results

| Setting | Number Acceptable (Out of 300) | | | |
|---|---|---|---|---|
| | Human 1 | Human 2 | Human Consensus | CSCS/Original Binomial $P$ |
| CSCS-proposed ($\beta = 0.25$) | 130 | 158 | 104 | $9.1 \times 10^{-8}$ |
| CSCS-proposed ($\beta = 1$) | 200 | 192 | 174 | 0.25 |
| Original headline | 223 | 233 | 197 | N/A |

**Significant Semantic Change.** For each headline, we considered all possible single-word mutations and picked the top according to the CSCS objective. Proposed mutations resulted in sentences that are qualitatively and quantitatively different than the original (**Figure 2**). CSCS often proposed word mutations that substantially change the part-of-speech (POS) structure. We quantified this observation by looking at the percentage of words in the mutated headline that had a different POS from the original headline. Using the NLTK POS tagger [10], the CSCS-proposed headline changed the POS of 16.7% of the words; using the FLAIR POS tagger [3], the median change was 14.3% of the words in the headline (**Table 1**). In contrast, the median POS change for the semantically-closest mutated headline (i.e., closest $\Delta\mathbf{z}[\tilde{x}_i]$) was 0% for both POS taggers (**Table 1**). Even when POS was not changed, CSCS proposed strikingly different word mutations, which we quantified using semantic similarity scores based on distance in the WordNet hierarchy [39, 28]. Specifically, for noun-to-noun and verb-to-verb changes, we selected the first WordNet synset corresponding to the depluralized or deconjugated version of the word. Across all these changes, the semantically-closest mutation had a median pathwise similarity of 0.14 and a median Wu-Palmer similarity [53] of 0.55 (both measures are between 0 and 1, inclusive, where 1 indicates high similarity, i.e., the same synset). In contrast, the CSCS-proposed mutation had a median pathwise similarity of 0.08 and median Wu-Palmer similarity of 0.24 (**Table 1**). Mean and standard deviation statistics, with similar trends, are also provided in **Table S1**. For both POS change and WordNet similarity, the difference between the CSCS-proposed and the semantically closest mutation are highly significant (two-sided independent $t$-test $P < 10^{-308}$). These results, supported by a qualitative examination of the changes (e.g., **Figure 2**), show that CSCS-mutated headlines are quite semantically different.

**Grammaticality Preservation.** We quantified grammaticality by asking human volunteers (12 in total) to provide grammatical acceptability labels. All humans were native English speakers with college degrees. Two humans were assigned to the same 150-headline text, blinded to the mutational status, and were asked to only evaluate grammaticality and not the content of the phrase, giving a binary "yes" or "no" label. Out of 300 original headlines, two humans provided a consensus "yes" grammatical label for 197 headlines (**Table 2**). The 300 corresponding CSCS-mutated headlines had 174 headlines with a consensus "yes" grammaticality; though lower, the number is within statistical error (two-sided binomial $P$-value of 0.25 compared with original). When we lowered $\beta$ from 1 to 0.25, thereby reducing the influence of $\hat{p}(x_i|\hat{\mathbf{z}}_i)$, consensus grammaticality of the 300 CSCS-mutated headlines dropped significantly to 104 (binomial $P = 9.1 \times 10^{-8}$; **Table 2**). These results suggest that by considering $\hat{p}(x_i|\hat{\mathbf{z}}_i)$, CSCS can preserve grammaticality. In general, CSCS of natural language produces intuitively satisfactory results and may be relevant to work in computational humor [52].

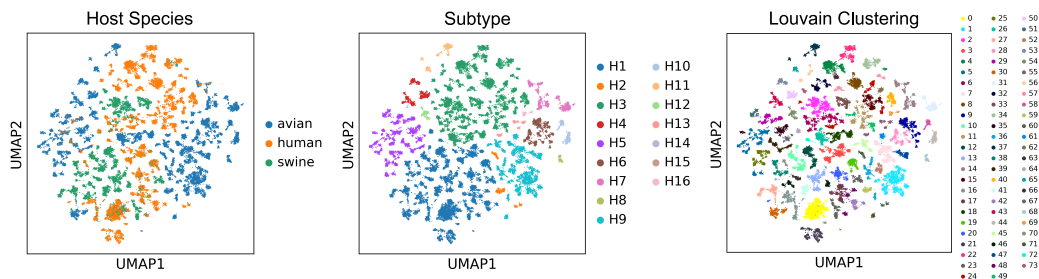

Figure 3: Semantic embedding space of influenza HA visualized in two-dimensions via UMAP [37] and colored by host species, subtype, or cluster labels from Louvain clustering [11].

## 4.2 Influenza

### 4.2.1 Language Model Training

**Training Data and Model Selection.** Our training data consists of 44,999 unique influenza A hemagglutinin (HA) amino acid sequences (around 550 residues in length) observed in animal hosts from 1908 through 2019. HA is a highly variable protein on the surface of influenza responsible for binding to host cells [24]. Since immunity to influenza is acquired by developing antibodies that bind and thereby neutralize HA, mutations to HA can lead to loss of immunity by reducing antibody binding affinity (i.e., immunological "escape") [30, 33]. Data was obtained from the NIAID Influenza Research Database (IRD) [55] through the web site at http://www.fludb.org (Appendix 6.1.2). These sequences were all obtained from animal hosts and thus, at least implicitly, encode viral viability and infectivity. We evaluated language model performance with a test set of held-out HA sequences where the first recorded date was before 1990 or after 2017, yielding a test set of 7,497 out of 44,999 sequences (about 17%). We again observed that a model with both an LSTM architecture and access to the full sequence context had the best train and test loss (Section 2.2.2).

**Semantically Meaningful Embedding Structure.** To improve our confidence that the embeddings are functionally meaningful, we leverage tools for unsupervised exploration of high-dimensional data. We trained our language model on the full IRD HA corpus, averaged $\hat{z}_i$ across all residues in each sequence (to enable comparison across variable length sequences), and visualized the resulting embedding in two dimensions with Uniform Manifold Approximation and Projection (UMAP) [37, 17, 7].This results in clear structure corresponding to influenza subtype and host species (**Figure 3**), which we quantify via unsupervised Louvain clustering [11]. Within each cluser, on average, 99.8% of sequences come from a single influenza subtype and 96.2% come from a single host species, indicating high correspondence between semantic structure and biologically important metadata.

### 4.2.2 Zero-Shot Escape Prediction with CSCS

**H3N2 Causal Escape Dataset.** We validate the ability for CSCS to prioritize escape mutations using an interventional dataset by Lee et al., who made all possible single-residue mutations to HA from the A/Perth/16/2009 (H3N2) strain and assessed which mutants preserve viral infectivity and induce escape [36]. To quantify escape, Lee et al. measured the overrepresentation of infectious viral sequences after immune selection by neutralizing human antibodies. These mutants therefore preserve infectivity and causally induce escape from neutralizing antibodies.

**CSCS Enrichment of Acquired Escapes.** Based on the language model trained over the full IRD HA corpus (Section 4.2.1), we computed $a'(\tilde{x}_i; \mathbf{x})$ for all possible single-residue mutations to the A/Perth/16/2009 HA sequence. We emphasize that *none of these mutants were present in the training corpus*. The mutants identified by CSCS are substantially enriched for experimentally-verified escapes from Lee et al. [36], e.g., 4 out of the top 5 hits were confirmed escapes (**Figure 4**). We quantified enrichment by computing the area under the curve (AUC) obtained by plotting acquired escape mutations versus total acquired mutations based on $a'(\tilde{x}_i; \mathbf{x})$, normalized by the maximum area to produce a score between 0 and 1, inclusive, where 0.5 indicates the expected value of random guessing. The AUC obtained by the full CSCS objective is 0.771, compared to 0.709 when acquiring

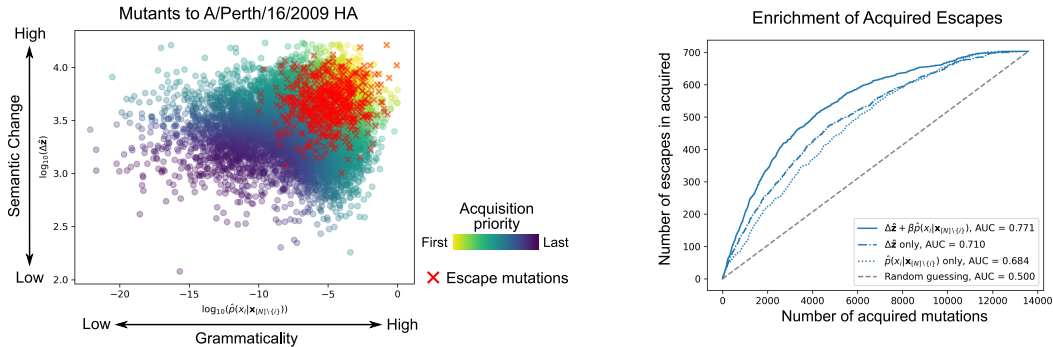

Figure 4: Left: Escape mutants (red Xs) to `A/Perth/16/2009` from Lee et al. [36] have high semantic change and grammaticality. Right: Escape mutants are substantially enriched in top CSCS-acquired mutants; see **Table 3**.

Table 3: Escape Prediction Results

|  | Normalized AUC | | |
| Model | Influenza H1 | Influenza H3 | HIV Env |
| --- | --- | --- | --- |
| MAFFT MSA | 0.697 | 0.598 | 0.523 |
| EVcouplings (independent) | 0.706 | 0.691 | 0.536 |
| EVcouplings (epistatic) | 0.726 | 0.687 | 0.552 |
| $\Delta \mathbf{z}[\tilde{x}_i]$ alone | 0.664 | 0.709 | 0.622 |
| $p(\tilde{x}_i|\mathbf{x})$ alone | 0.820 | 0.684 | 0.667 |
| **CSCS** ($\Delta \mathbf{z}[\tilde{x}_i]$ and $p(\tilde{x}_i|\mathbf{x})$) | **0.834** | **0.771** | **0.692** |

solely based on semantic change and 0.684 when acquiring solely based on grammaticality (**Figure 4**; **Table 3**), indicating that both are informative for determining escape. We obtained these results without direct supervision or explicit escape training data.

**Benchmark of Existing Approaches.** Though to our knowledge no previous method has been explicitly designed for escape prediction, we compare with standard viral fitness model strategies that are the closest to our unsupervised problem setting. The first strategy performs MSA of the viral sequence corpus and acquired escapes are simply those with the highest observed mutational frequency [12, 31, 4, 21]; our two benchmark methods that leverage this strategy are MAFFT MSA [29] and EVcouplings independent [26] (see Appendix 6.2.2 for more information). The second strategy also requires MSA followed by parameter estimation in a Potts model [27, 26], which incorporates pairwise residue information; we use the EVcouplings epistatic model that implements this approach, which is described in greater detail in Appendix 6.2.2. For influenza, we observed consistently higher AUCs obtained by CSCS over all benchmark methods (**Table 3**), noting that these methods were not specifically designed for viral escape prediction. We also tested pretrained protein sequence embedding models [9, 45, 5], not trained on viral corpuses, to see if their representations automatically transferred to viral escape prediction (Appendix 6.2.3), but this was not the case (**Table S2**), indicating that specific viral training data greatly improves escape prediction.

**CSCS of H1N1 Viral Mutations.** We evaluated CSCS on HA from another flu strain, `A/WSN/1933`, from a different HA subtype (H1 instead of H3) for which causal escape mutations were also determined by the same experimental procedure above, albeit with a more limited set of neutralizing antibodies [19]. Using the same language model trained on the IRD corpus, we ranked all possible single-residue mutations of `A/WSN/1933` HA, $\tilde{x}_i$, based on $a'(\tilde{x}_i; \mathbf{x})$. We again found substantial enrichment of escapes (observed in [19]) in the top mutations; the normalized AUC of acquired escape mutations versus total acquired mutations was 0.834 (**Table 3**). We note that none of these mutated sequences were present in the training data. In contrast, other approaches had lower enrichment of acquire escape mutants (normalized AUC $\leq$ 0.726; **Tables 3** and **S2**). Though similar causal escape

data is not available for other influenza strains, this additional validation increases our confidence that escape prediction with CSCS generalizes across strains.

## 4.3 HIV

**Setup and Training Data.** To assess generality to other viral proteins, we analyzed the HIV-1 Envelope (Env) protein, which, like influenza HA, is responsible for binding and entering host cells and is also targeted by antibodies [6]. Env is larger than influenza HA (about 850 residues compared to around 550) and more readily escapes immune selection due to viral mutation, even within the same host [47]. We train our language model on 60,857 unique Env sequences from the Los Alamos National Laboratory (LANL) HIV database (Appendix 6.1.3) [23]. We used the same language model architecture as in the influenza HA experiments. We again observed functionally-meaningful patterns when visualizing the semantic embeddings of Env sequences (**Figure S1**).

**Zero-Shot Escape Prediction with CSCS.** We applied CSCS to a dataset quantifying the infectivity and escape potential of all single-residue mutations to Env from the BG505.T332N strain of HIV, using a similar experimental procedure as that for HA from the two influenza strains described above [18]. We ranked all single-residue mutations $\tilde{x}_i$ of BG505.T332N by the CSCS objective $a'(\tilde{x}_i; \mathbf{x})$. We again observed enrichment of escape mutations when acquiring based on both semantic change and grammaticality, though with a weaker enrichment than observed for influenza HA (normalized AUC = 0.692; **Table 3**), suggesting that the semantic complexity of HIV Env might be more difficult to model with existing training data. However, CSCS escape prediction performance still exceeds that of other models (normalized AUC $\leq$ 0.574; **Tables 3** and **S2**).

## 5 Discussion

Here we show that a learning-based, distributional approach to modeling viral sequence achieves unprecedented insight into evolution and escape, suggesting a timely and important direction for the machine learning community. Excitingly, we demonstrate that the distributional hypothesis is a productive assumption for analysis of viral variation. This is not obvious, since it may be possible for non-causal mutations to widely co-occur with causal escape mutations [32], but our results suggest that many of the mutations that alter distributional structure are also causal escape mutations (perhaps due to pressure on viral sequences to maintain both diversity and economy, thereby diminishing the importance of non-causal mutants).

The CSCS problem in general is useful for any domain in which substantial functional change is desirable but the feature changes are limited or constrained. For example, in exploring differences in human-versus-machine perception, it may be desirable to generate entities that are perceived as similar by humans but as vastly different by algorithms, or vice versa. Though we focus on zero-shot, unsupervised escape prediction, some supervision could be useful in improving performance (e.g., learning $\beta$ from a handful of examples).

A broader problem is in modeling other changes aside from mutations, like insertions and deletions, or more complex sequence changes. CSCS that accommodates insertions and deletions (about four times rarer than mutations in viruses [48]) could likewise model semantic change as a shift in the embedding space and grammaticality as some function of an emitted language model probability. While single-token changes allow for interpretability and efficiency, CSCS could be extended to multi-token changes (e.g., by combining the individual mutational probabilities to approximate the joint probability), though the search problem then becomes combinatorial. It may also be possible to evolve a sequence over multiple timesteps, each with a new single-token change, to produce complex sequence designs.

## Broader Impact

We hope that this work leads to broad positive impact by (1) encouraging those in the machine learning community to contribute to understanding and combatting viruses (and infectious disease more broadly) and by (2) providing state-of-the-art prediction of how viruses can mutate around neutralization, which could be useful as part of rational design of vaccines or therapies. *In silico* models of how mutation leads to pathogenesis might help reduce both the resources and risks

associated with experimentally characterizing viral mutants. A primary goal of infectious disease research in general is to mitigate and prevent pandemic disease events among the global human population, which lead to widespread mortality, suffering, and economic disruption.

In computationally predicting mutations that induce escape or improve viral fitness, misuse could potentially take the form of using such methods to increase the pathogenicity of an existing viral strain. Experimental biologists, policy makers, and ethicists have already devoted and continue to devote a substantial amount of consideration to the ethics of such "gain-of-function" research (GOFR) [1, 49, 50]. As computational biologists become part of the GOFR conversation, attention to ethics is paramount and the scientific community should continue to preserve and strengthen the existing combination of experimental and policy safeguards.

Work in this area should continue to rely on direct experimental validation of computational prediction so that any system failures can be identified and corrected. Global viral surveillance already benefits from international cooperation through entities like the World Health Organization and collaborations like the Global Virome Project [13], and both the IRD and LANL HIV databases already have substantial global coverage across six continents [55, 23]. Preventing datasets from bias toward certain geographies or human populations underscores the already high priority given to viral monitoring at a global scale.

## Acknowledgments and Disclosure of Funding

We thank Alejandro Balazs, Owen Leddy, Adam Lerer, Allen Lin, Adam Nitido, Uma Roy, and Aaron Schmidt for helpful discussions. We thank Steven Chun, Benjamin DeMeo, Ashwin Narayan, An Nguyen, Sarah Nyquist, and Alexander Wu for assistance with the manuscript. B.H. and E.Z. are partially funded by NIH grant R01 GM081871 (to B.A.B.). B.H. is partially funded by the Department of Defense (DoD) through the National Defense Science and Engineering Graduate Fellowship (NDSEG). E.Z. is partially funded by the National Science Foundation (NSF) Graduate Research Fellowship Program (GRFP). B.D.B. acknowledges funding from the Ragon Institute of MGH, MIT, and Harvard; MIT Biological Engineering; and NIH grant R01 A1022553.

## Footnotes

[1]Code at https://github.com/brianhie/mutational-semantics-neurips2020.

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
