[Supplementary Material]

# 6 Appendix

## 6.1 Additional dataset details

### 6.1.1 Headline dataset

Preprocessed headlines (stripped of punctuation, space-delimited, and lower-cased) from the Australian Broadcasting Corporation (early-2013 through the end of 2019) were obtained from `https://www.kaggle.com/therohk/million-headlines`.

### 6.1.2 Flu IRD dataset

Influenza HA amino acid sequences were downloaded from the "Protein Sequence Search" section of `https://www.fludb.org`. We only considered complete HA sequences from virus type A, but did not filter based on subtype, strain, date, host, geography, or country.

### 6.1.3 HIV LANL dataset

Sequences were downloaded from the "Sequence Search Interface" at `https://www.hiv.lanl.gov`. All complete HIV-1 Env sequences were downloaded, excluding sequences that the database had labeled as "problematic." To ensure that our sequences corresponded to complete viral haplotypes, we only considered sequences that had length between 800 and 900 amino acid residues, inclusive.

## 6.2 Additional baseline method details

We benchmark our escape prediction experiments against models that try to estimate the evolutionary fitness of a viral protein based on some assumptions. Notably, viral fitness models are not equivalent to escape prediction, since mutations that preserve fitness may be neutral with respect to escape (fitness models better correspond to the "grammaticality" term in CSCS). However, in the absence of existing unsupervised models that are directly built to perform unsupervised escape prediction, viral fitness models are the most related that attempt to solve a conceptually close problem.

### 6.2.1 Alignment-based frequency fitness model

This baseline model for viral fitness assumes that higher mutational frequencies in a corpus correspond to higher fitness and that residue-level fitness information is independent across the viral sequence; this fitness model is widely adopted due to its simplicity [12, 31, 4, 21].

We first perform MSA with the MAFFT software package (version 7.453) within the respective corpuses (influenza or HIV sequences). After sequence alignment was performed, we considered each position in the viral sequence of interest (influenza strains `A/Perth/16/2009` or `A/WSN/1933`, or HIV strain `BG505.T332N`). At a given position, we computed the frequency of other amino acids that were aligned to that position across all other sequences in the corpus. Sequences were acquired based on the highest observed frequencies across all possible single-residue mutations.

For influenza, we found that performance (in terms of normalized AUC) improved when restricting sequence alignment to the corresponding subtype (H1 sequences for `A/WSN/1933` and H3 sequences for `A/Perth/16/2009`) For HIV, we found that performance improved when only restricting alignments to the local neighborhood of `BG505.T332N`, defined by sequences that differ by a maximum of 15 residues. In general, we found that sequence alignment is dramatically affected by the sequences that are included in the corpus. For a best-case comparison, we report the highest performance over different sequence inclusion strategies.

We also used a conceptually similar implementation of this strategy provided by the EVcouplings pipeline [26] (`https://github.com/debbiemarkslab/EVcouplings`) using default parameters. We trained the EVcouplings independent model on the same corpus of viral sequences used to train our language models.

### 6.2.2 Alignment-based Potts model

A common critique of the above strategy for modelling viral fitness is that the independence assumption is limiting. Biologically, two residues can co-evolve, especially if they are physically and

biochemically related in the three-dimensional structure of the protein, a phenomenon referred to as "epistasis." A solution is to incorporate pairwise residue information by learning a probabilistic model in which each residue position corresponds to a random variable and pairwise potentials can encode epistatic relationships.

Hopf et al. learned such a model based on a Potts model formulation; we describe the general formulation here and leave implementation details to the original paper [27]. Given a sequence $\mathbf{x} = (x_1, x_2, \ldots, x_N)$ where $x_i$ comes from an alphabet $\mathcal{X}$ that is the set of all amino acids and a gap character, the model assigns an energy score to each sequence as

$$E(\mathbf{x}; \mathbf{h}, \mathbf{J}) \triangleq \sum_{i=1}^{N} h_i x_i + \sum_{i=1}^{N} \sum_{j=i+1}^{N} J_{ij} x_i x_j.$$

This term is scaled to be a valid probability distribution

$$p(\mathbf{x}; \mathbf{h}, \mathbf{J}) = \frac{1}{Z} \exp \left\{ -E(\mathbf{x}; \mathbf{h}, \mathbf{J}) \right\}$$

where $Z = \sum_{\mathbf{x}'} \exp \left\{ -E(\mathbf{x}'; \mathbf{h}, \mathbf{J}) \right\}$. The parameters are learned by a maximum likelihood procedure using a number of critical heuristics that Hopf et al. use to allow for efficient inference and parameter regularization [27, 26]. We use the pipeline provided by Hopf et al. at `https://github.com/debbiemarkslab/EVcouplings` with default parameters. We trained the EVcouplings epistatic model on the same corpus of viral sequences used to train our language models.

### 6.2.3 Pretrained sequence embedding models

We tested if the sequence embeddings produced by models trained on generic protein sequence corpuses [9, 45, 5] would be informative with respect to escape. We used the pretrained transformer model from Rao et al. [45] and the pretrained UniRep model from Alley et al. [5], both obtained through `https://github.com/songlab-cal/tape`. We used the pretrained model with full soft symmetric alignment and protein structure information from Bepler et al. [9], available through `https://github.com/tbepler/protein-sequence-embedding-iclr2019`. Rather than training exclusively on a large viral sequence corpus, as we did, these methods trained on corpuses containing generic protein sequences.

Each single-residue escape mutant was embedded using the pretrained model and mutant sequences were acquired based on the largest changes to the embedding based on the $\ell_1$-distance. The results are provided in **Table S2**.

### 6.3 Additional experimental details

### 6.3.1 Language model hyperparameter selection

We performed a small-scale grid search using categorical cross entropy loss after 20 training epochs on the headline and influenza datasets to select the language model architecture and hyperparameters based on a random 80%/20% cross-validation split of the training set. Hyperparameter ranges were influenced by previous applications of recurrent architectures to protein sequence representation learning [9]. We tested hidden unit dimensions of 128, 256, and 512. We tested architectures with one or two hidden layers. We tested three hidden-layer architectures: a densely connected neural network with access to both left and right sequence contexts, an LSTM with access to only the left context, and a BiLSTM with access to both left and right sequence contexts. We tested two Adam learning rates (0.01 and 0.001). All other architecture details described in Section 2.2.2 were fixed to reasonable defaults. In total, we tested 36 conditions and ultimately used a BiLSTM architecture with two hidden layers of 512 hidden units each, with an Adam learning rate of 0.001. We used the same architecture for all experiments. In general, we noted that increasing model capacity only served to improve performance.

### 6.3.2 Headline semantic change quantification

POS tagging was done using the English `pos_tag()` function with default parameters from the `nltk` Python package (`https://www.nltk.org`) and separately using the default POS `SequenceTagger` from `flair` (`https://github.com/flairNLP/flair`).

CSCS-mutated words were compared to the original word based on WordNet synset similarities. We only considered words where the POS (labeled by `nltk`) was preserved, where the POS was a noun or a verb (i.e., `NN`, `NNS`, or `VB`), and where the depluralized or deconjugated word was present in `nltk`'s WordNet. We used the `pattern` Python package (`https://github.com/clips/pattern`) to depluralize words or to conjugate verbs into the infinitive form.

### 6.3.3 Computational resources

Training on the influenza HA dataset requires approximately a week of training and around three hours to evaluate all possible single escape sequences. On our largest dataset (HIV Env), our training implementation finished within 2.5 weeks and escape prediction inference requires eight hours. Models were trained with an Nvidia Tesla V100 PCIe 32GB GPU. Experiments were run with Python 3.7 on Ubuntu 18.04.

### 6.3.4 Code and data availability

Code and datasets used in this paper's experiments has been made available as supplementary data.

Table S1: Additional Headline Semantic Change Results

| Setting | Mean $\pm$ S.Dev. % POS Change | | Mean $\pm$ S.Dev. WordNet Similarity | |
|---|---|---|---|---|
| | NLTK | FLAIR | Pathwise | Wu-Palmer |
| Semantically closest (smallest $\Delta\mathbf{z}[\tilde{x}_i]$) | $8.40\% \pm 13.3\%$ | $5.64\% \pm 10.5\%$ | $0.266 \pm 0.280$ | $0.536 \pm 0.298$ |
| CSCS-proposed (highest $a'(\tilde{x}_i; \mathbf{x})$) | $18.9\% \pm 15.3\%$ | $15.5\% \pm 14.3\%$ | $0.0833 \pm 0.0756$ | $0.235 \pm 0.145$ |

Table S2: Additional Escape Prediction Results (pretrained sequence embeddings)

| Model | Normalized AUC | | |
|---|---|---|---|
| | Influenza H1 | Influenza H3 | HIV Env |
| Alley et al. pretrained $\Delta\mathbf{z}$ [5] | 0.482 | 0.452 | 0.534 |
| Bepler et al. pretrained $\Delta\mathbf{z}$ [9] | 0.660 | 0.644 | 0.561 |
| Rao et al. pretrained $\Delta\mathbf{z}$ [45] | 0.584 | 0.526 | 0.574 |
| **CSCS** ($\Delta\mathbf{z}[\tilde{x}_i]$ and $p(\tilde{x}_i|\mathbf{x})$) | **0.834** | **0.771** | **0.692** |

Figure S1: UMAP visualization of unique HIV Env sequences colored by subtype. Large, dominating clusters corresponding to B, C, and AE subtypes may be due to the lack of vaccine pressure on HIV, compared to influenza.