[Reviews · NeurIPS 2020]

Review 1

Summary and Contributions: This paper proposes a novel formulation of viral escape prediction (essentially predicting virality from protein sequence) as a “constrained semantic change search” (CSCS) problem, in which we seek the mutation(s) that change semantics while still preserving grammaticality. Notions of semantics and grammaticality are borrowed from NLP, where the former is captured by contextual and co-occurrence patterns of symbols (either words or protein residues). The paper then shows how to adopt the embeddings learned by language models to perform CSCS, and demonstrates good performance empirically on both viral escape prediction (on three different viral protein targets) and a task of modifying news headlines to be as semantically different as possible while still grammatically valid.

Strengths: The paper proposes a novel formulation of the difficult biological task of viral escape prediction, which allows one to leverage existing language model architectures and training procedures. The paper also demonstrates good empirical performance of their proposed method, compared to some simple baselines based on multiple sequence alignments. Line 182-183, 192-194: Hugely appreciate the use of statistical tests to quantify improvements. It might be nice to mark significant differences directly in Table 1.

Weaknesses: There are a few weaknesses that might be helpful to address (also see comment on Correctness): clarification of the notion of “grammaticality” for this problem; further connections to similar approaches for protein modeling that could be considered; the slightly ad hoc nature of the CSCS objective; and the fact that comparisons made to similarly high-capacity deep unsupervised models in the Appendix did not use viral data. These are explained in further detail below: Appropriateness of “grammaticality” for the viral immunological escape problem: I appreciate the trend of using massive amounts of unsupervised data to circumvent the difficulty of obtaining fitness measurements for biological sequences, which this work also advances. However, there is some degree of implicit supervision involved here, in that the amino acid sequences used (described in Lines 199-200) are explicitly from infectious viruses (rather than somehow being neutral/benign). It’s also not clear that casting this observation as merely an issue of “grammaticality” makes sense: if a “grammatically correct” sequence is one that belongs to an infectious virus, what’s the difference between grammaticality and semantics (which are also supposed to capture what makes a sequence infectious)? Perhaps one could claim that grammaticality in this context has to do with whether the protein is “valid”, in the sense that it folds or is stable, but this is not explained, and does not absolve the first point that all the sequence data come from infectious viruses (rather than, for example, all valid protein variants that fold/are stable, which would allow for a much clearer distinction between grammaticality and semantics). Lines 128-133, 136-138: This is a nice point, and one I think it would be helpful to amplify a bit more in comparing to related biology work. There is a whole line of work in computational biology using either 1) generative models of unlabeled evolutionary data (natural homologs) to identify the effects of mutations in proteins (e.g., Hopf et al. 2017, Riesselman et al. 2018), or 2) unlabeled and unaligned proteins to aid learning models of fitness functions (from which one could extract coefficients related to the effects of particular mutations) (e.g., Alley et al. 2019 cited in your paper, Biswas et al. 2020). It’s helpful for the reader to realize that such precedents may not be applicable, due to the unique nature of viral evolution. On the other hand, it would also be clarifying to point out that for other biological domains, such as proteins and other genomes that do not evolve as quickly, such existing methods may be sufficient to solve the high-level problem posed in this paper. It would also be interesting to consider the applicability of 2). Though direct measurements of viral fitness are limited, as described, there do exist datasets with such measurements for particular viral strains (such as those used in this paper). One could imagine an approach very similar to that of Alley et al. 2019, wherein unsupervised learning on a large unlabeled dataset of viral genomes is used to learn a feature space for a supervised regression model (on top of even extremely small numbers of labeled genomes, as in Biswas et al. 2020). The coefficients of that regression model could then also be used to rank mutations. The benefit of the CSCS approach, of course, is that no labeled data is needed whatsoever, but it’s interesting and instructive nonetheless to consider these comparisons. Lines 57-58: Though the objective a(\tilde{x}_i ; x) (it would also help to label this with an equation number, even if just for review purposes) has an intuitive form, the appeal to Bayesian optimization as a motivation doesn’t make much sense. Adding a weighted uncertainty term in BO constructs a confidence bound on the value of the underlying function, whereas here, the weighted term has nothing to do with uncertainty and actually captures how grammatically wrong a mutation is. The usage here is much more akin to the Lagrangian of a constrained optimization problem, where \beta can have the interpretation of how difficult grammatical constraints are to satisfy. It would improve the motivations and development of the method to ground this particular form in a more principled way, whether with this interpretation or another. Appendix Section 6.2.3: The comparison to these methods is much appreciated, and (subject to space constraints) it would be worth trying to show them in the main text. Particularly for the machine learning community, these are more convincing points of comparison than the MSA-based methods. However, the obvious flaw in these comparisons (as noted in Line 250) is that the competitor models were pre-trained on entirely different *non-viral* proteins, unlike the results shown for CSCS, so it’s not surprising at all that the competitor models probably learned patterns of residue covariation that are totally irrelevant to viral evolution and selective pressures. Consequently, all the methods compared against are “unfair” in one way: the MSA-based competitors fit generative models of relatively low expressive capacity (or at least, impose a very particular form), but are fit to the relevant viral data, whereas the deep unsupervised competitors in Appendix 6.2.3 are high-capacity, but are not trained on the relevant viral data. References T. A. Hopf, et al., Mutation effects predicted from sequence co-variation. Nat. Biotechnol. 35, 128–135 (2017). A. J. Riesselman, J. B. Ingraham, D. S. Marks, Deep generative models of genetic variation capture the effects of mutations. Nat. Methods 15, 816–822 (2018). S. Biswas, G. Khimulya, E. C. Alley, K. M. Esvelt, G. M. Church, Low-N protein engineering with data-efficient deep learning. bioRxiv, 2020.01.23.917682 (2020).

Correctness: Lines 157-163: Hopefully this is a clarity issue, but was there no validation set? That is, were model architecture and hyperparameters were chosen using the same test set that semantic change results are reported on (i.e. the “test set” in Line 157 is the same set from which “each headline” in Line 164 was drawn)? If so, this is incorrect experimental procedure that invalidates the results reported in Table 1, Figure 2, and Table 2.

Clarity: Line 60: It would be helpful to state explicitly here whether \beta is a parameter that is learned, or a hyperparameter that is set. Section 2.1: It would be helpful to state explicitly, much earlier than Line 70, that the parameters one needs to learn are both the embedding function f_s and the distribution p(\tilde{x}_i \mid x) that encodes soft grammaticality constraints. It was not clear from the language in Section 2.1 that CSCS itself does not explicitly involve learning these functions/distributions, so it would be helpful to clarify that, by noting e.g. that such functions/distributions are already given or induced by language models. It should be made clear as early as possible that the novelty of CSCS is in leveraging these models in a new way, not in proposing another way of training them. Line 70: The phrase “over all i \in [N]” makes it seem like these functions/distributions are functions of integers i \in [N] (it’s also unclear from this language if there are N different encoding functions, one for each position i). Something like “... function \hat{f}_s, where, for all i \in [N], …” would be clearer. (The following equations should also have numbers, even if only for review purposes.) Lines 71-72: It seems that such embeddings z are independent of the position i. If so, it might be helpful to explicitly state that for clarity. Equations after line 74: Shouldn’t \hat{p}(\tilde{x}_i \mid \hat{z}_i) be conditioned on \hat{z} rather than \hat{z}_i, to capture the embedding of the whole sequence (and to be analogous to p(\tilde{x}_i \mid x)? If not, why not? Also, is there a particular reason the \ell_1 rather than Euclidean norm is used, or was it just the one that worked better empirically? Section 2.2.3 The usage of “acquisition” here is slightly confusing. Is it being used in a particular technical sense (e.g. what does “acquiring mutations” mean)? This also induces ambiguity in what it means to “calibrate \beta”; presumably this means to pick the value of \beta that provides the best performance, but it hasn’t been defined/is unclear what constitutes good performance in this setting. (If this is what “calibrate” is being used for, I might suggest a different term to avoid confusion with the sense of calibrating a probabilistic model.)

Relation to Prior Work: This paper draws connections to work spanning both the machine learning and computational biology literatures, and does so relatively thoroughly. See second paragraph of comments on Weaknesses, however, for potential further discussion of connections/comparisons to work on protein modeling.

Reproducibility: Yes

Additional Feedback: Update after author response/discussion: I realize I had initially failed to fully appreciate the paper's contribution in identifying a new and useful NLP task. It’s significant in and of itself not only because it proposes a new language modeling task, but also because it is rare for creative advancement on a biological problem to inspire innovation in NLP (whereas the other direction, adapting existing NLP techniques for e.g. protein problems, has recently become more common). The authors also did a great job with the response, particularly in clarifying their problem set-up and adding comparisons to evolution-based methods that are more commonplace for non-viral proteins, e.g. EVcouplings. I’m happy now to increase my score.


Review 2

Summary and Contributions: The paper introduces a new NLP “downstream task” called “constrained semantic change search” (CSCS). It consists in finding a single change in a sequence that will have the most impact on the semantics while still being grammatical. The authors define the task formally and propose a simple surrogate function to optimize in order to solve the task. The authors experiment on news headlines as a sanity check to see if their assumptions and algorithm are valid. They then proceed to an actual useful application of their new task, that is to find escape mutations in viruses. Their experiments show that NLP tools such as embedding spaces and RNN can also be applied successfully in other domain such as virology.

Strengths: The authors identify a new downstream task with an application to virology, which could have a large impact in the scientific community. The paper could be a major step to bring virology closer to machine learning. I can see many extensions and future applications of this work. The theoretical part is simple and easy to understand, and the experimentations convince that the authors are onto something. The zero-shot learning makes their findings even more strong.

Weaknesses: The authors try to justify their theoretical setting with claims that are not entirely correct (see the Correctness section), but the mathematics seem sound and the experiments support it. A limitation of the paper is that it handles only a single mutation at a time and not insertions nor deletions.

Correctness: The paper makes some assumptions that are not completely correct. The authors assume that “closeness in embedding space corresponds to semantic similiarity” (line 54). However, antonyms can be close in this space. One could argue that when looking for the most semantic change in a sequence, choosing the antonym of a word could sometimes be a good choice, but the proposed method would ignore those words. On the other hand, I suppose that the concept of “antonyms” is not present in viral proteins, so it should not matter too much in that case. Furthermore, the author assumes that the probability of a token returned by the language model provides grammaticality. However, it encodes much more than the grammaticality, since it tells if the token is probable in this context or not. The author should make an explicit remark that p is a surrogate for the grammaticality, or simply refer to this term as the “plausibility” or “likelihood” of the mutation. For this reason, I feel the grammatical acceptability experiments (Table 2), while being interesting, is not a good evaluation of the task for the reasons explained above. In Table 1, the authors compare the results obtained by using the highest value of a’ versus the lowest Delta z. However, it would have been fairer in my opinion to compare versus the lowest Delta z while maximizing “grammaticality”, i.e. inserting a negative sign before the “rank(Delta z[x_i])” in the equation below line 104. It would also have been a good addition to compare different values of beta to illustrate the trade-off that this parameter can have. Moreover, the LSTM architecture is very basic. I expect that more recent architectures with contextualized embeddings, such as ELMo or RoBERTa, could be very profitable for the task at hand. However, since this is apparently the first work on this topic, it is excusable to go with a simple baseline to limit extraneous factors. Aside from this, the experiments seems well done and are convincing.

Clarity: The paper is very well written: it is easy to follow and I did not catch many typos or errors. Sections are well defined, and it is clear where the paper is going. One downside I would underline is the lack of background content relating to viral proteins. Since the NeurIPS community is likely to know nothing about this field, it would be beneficial to at least explain what is the “alphabet X” that is used for proteins. Furthermore, it is not known until section 4.3 on HIV what is the mean length of influenza proteins. I think this information should be present in section 4.2.1. I had to read multiple times and think to understand the sentence starting at line 232. Maybe the author could rework the explanation of the operation that is done. The figures are well presented, useful and interesting.

Relation to Prior Work: The authors discuss related work, and it is clear that the paper is not a simple increment on existing work. The CSCS task and its application to viral proteins in a zero-shot setting to identify escape mutations is novel to me.

Reproducibility: Yes

Additional Feedback: Here are some other minor observations that could be improved. 1. At line 50 and other places, you speak about an “alphabet X”. Maybe the terminology “vocabulary” would be more appropriate, since “alphabet” can also refer to characters. 2. Below line 74, the definition symbol is :=, but everywhere else a triangle above the equal symbol is used. 3. At line 92, it is said that the “Null character prepadding was used to handle variable-length sequences”. A LSTM is a RNN and is made to handle sequences of variable length. Why do you need to pad them with a null character? 4. At line 119, I would argue that the sentence “Typically, natural language applications are mostly concerned with searching for entities based on semantic closeness.” is false. NLP is a vast domain, and many tasks are not related to semantic closeness. Maybe you meant “named-entity recognition”, but even then such a statement would require justification. 5. The modification of the objective a to the rank-based objective a’ seems a little bit artificial. A comparison of both objectives (at least in the Appendix) would have been beneficial. Response to author feedback: Your feedback was satisfactory and answered well to concerns of other reviewers. After reading the other reviews, I could also add a suggestion to include more references to standard assumptions and hypotheses related to the NLP field, since some readers in the field of virology may be unfamiliar with these.


Review 3

Summary and Contributions: The authors tackle the problem of identifying mutations in viruses that might result in viruses escaping antiviral treatments or immunity. The authors propose that this problem is equivalent to identifying word changes to sentences that induce a large change in semantics but retain grammaticality. The proposed approach, Constrained Semantic Change Search (CSCS), learns a semantic embedding of the words/amino acids using the sequence (sentences / protein sequences) context. This is achieved by first learning a latent representation using the biLSTM language models. This model is assumed to capture the full sequence context of a word such that the word is conditionally independent of the context given the latent representation. The authors then present a scoring / ranking scheme that is a weighted combination of semantic changes, the distance in the latent space and the grammaticality, the probability of observing the change given the context captured by the latent representation. A key aspect of the framework is that it is unsupervised - the context alone is used to learn the latent representation and does not require a training set of known grammatically accurate semantic changes or mutations resulting in higher viral infectivity. The CSCS framework is first tested on sentences using news headlines. The evaluation performance is laid out well with tests to measure semantics, grammaticality and the importance of using both aspects in inducing changes. The authors then apply this to learn influenza virus mutations. The results demonstrate that the embedding derived by CSCS is semantically correct based on host species and viral strains. A prior experimental dataset that measures viral escape is used to asses the output of CSCS - CSCS outperforms existing methods as measured by auROC.

Strengths: * The problem formulation and framework using language models is a strong setup in my opinion. The ranking scheme effectively balances the trade-off between semantic change and grammaticality in the same latent space. * The unsupervised nature of the problem - the latent representation are derived from sequence context, is a key strength given the sparse availability of expensive training data for measure viral infectivity and escape * As the authors point, the use of language models to predict viral escape is novel and could be significant if practical predictions can inform experimental design

Weaknesses: * The main issue I see is that the authors do not provide bounds or guarantees or commentary on how the model architecture can ensure that the latent representation satisfies the key property of x_i being conditionally independent of the context given the latent representation value z_i. * I am also not convinced that auROC is exactly the right metric to use to asses the performance on viral escape data. I believe it is important to consider a continuous measure of viral infectivity rather than whether a mutation can lead to immune escape. * The practical implications of CSCS framework on follow up experimental testing are not clearly laid out - how would the ranking scheme influence follow up experimental design? What is the number of candidates to be tested given the variable performance rates in different viruses?I believe this is important given the stated goal. * While the framework can provide insight about viral escape, I am not sure if this directly informs virus evolution as claimed by the authors since the mutations are not directly modeled

Correctness: I do not believe the claims or the method are in correct but the evaluation and practical implications can be improved.

Clarity: Yes, the paper is well written and is easy to follow.

Relation to Prior Work: Yes, I think so.

Reproducibility: Yes

Additional Feedback:


Review 4

Summary and Contributions: The authors propose a method, constrained semantic change search, which seeks "grammatically valid" changes to a sequences that induce the largest "semantic change". The motivating application for this work comes from virology. The goal is to find viral sequences which are grammatically valid but induce large "semantic" changes. Here, a "semantic" change corresponds to a mutation which has significant effect on the ability of the virus to evade a hosts immune response. The authors demonstrate a proof of concept on a text application as well as on real viral sequence data.

Strengths: Conceptually I think the work is quite interesting. The authors draw from the NLP literature to develop models to predict viral escape. The demonstration on text data provides some nice intuition. Also, the empirical performance (especially Fig. 4) is quite compelling.

Weaknesses: I think one of the main weaknesses for me is a lack of mathematical or theoretical detail. I understand that this is a more applied work but I don't have a sense of exactly how the method works or what choices were made. I was originally confused about how the semantic change was computed. If I understand correctly (bottom of page 2) the authors compute the embedding for ever subsequence of length N-1, and concatenate them together to create z_hat? And each Zhat_i is 20dimensional, so this is an zhat is N x 20 matrix? If so this seem a bit odd and ad hoc: since all embeddings are based on sequences of N-1 but the real semantic change should operate on embeddings for sequences of length N. Why is the L1 norm used? Does zhat include z_i? if so, the semantic change for that column should be 0 right? Assuming p(x) and Z are inferred for every sequence. Do the authors just use an exhaustive search to identify mutations that cause the largest semantic change? Is this feasible? The objective function based on a linear combination of grammaticality and semantic difference seems arbitrary. How sensitive are the results to the choice of beta? It seems like the authors only compare 2 values (0.25 an 1). What about values greater than 1? UPDATE: The authors did a nice job of addressing most of the above comments, although I'm still left a bit unsatisfied about the choice of beta. I can imagine the effectiveness of this approach can ve quite sensitive to this choice depending on the problem. On aggregate I think this is a nice paper, but I'm not compelled to increase my score further.

Correctness: Yes.

Clarity: The paper is generally clear and well written though details are very sparse which makes it challenging to full understand the work.

Relation to Prior Work: Yes, the authors include a nice section on the relationship to prior work.

Reproducibility: No

Additional Feedback:

[Author Response · NeurIPS 2020]

We are grateful to all reviewers for their high-quality consideration of our manuscript. There are some key clarifications
we want to make regarding the nature of the viral escape problem, and some additional analyses we did during the
intervening review period. While our comments are organized by reviewer, we hope the totality of our response will be
useful to all. We will also link to a public GitHub for reproducibility in the final manuscript version.

**Reviewer 1:** Thanks for noting our novelty and empirical performance. • "*Appropriateness of 'grammaticality'*": We
wanted to clarify a key point about viral escape (and apologize for our lack of clarity), which we believe addresses your
concern. To escape from human immunity, not only does a mutation need to preserve infectivity, but it *also* must be
functionally/antigenically altered enough so that antibody recognition no longer works. So, the fact that our training
data comes solely from infectious virus, which would be highly probable (or "grammatical") sequences under our
language model (LM), is a key feature of our approach. We intuit that it is "grammaticality" that encodes infectivity,
while it is "semantic change" that encodes the functional/antigenic alteration essential to immune escape (rather than,
e.g., a mutation that preserves infectivity but is functionally neutral and therefore does not affect antibody recognition).
• "*consider these comparisons*": Regarding points about comparison/benchmarking on viral finetuned embedding
models and generative models of evolutionary data, during the review period, we have auspiciously performed new
benchmarks that we can hopefully add to the paper. We newly benchmark with EVcouplings (a refined version of the
method in Hopf et al., *NBT*, 2017) on the same viral training dataset as our model and we were also able to update
Bepler/TAPE results as well. Among these, the highest AUC for H3 is EVcouplings independent with 0.691, for H1 is
EVcouplings epistatic with 0.726, and for Env is TAPE with 0.574, which are all lower than the CSCS results with our
LM. Importantly, however, we note that, fundamentally, CSCS is presented in generality here so these methods are
not strictly "competitor methods" in the sense that, if one were to work better, it would still be incorporable within
the CSCS framework. • "*slightly ad hoc nature of the CSCS objective*": One nice thing about the CSCS objective is
that it has a straightforward interpretation and its simplicity (combined with our empirical results) illustrates a rather
direct connection between our modeling intuition and nature, and we can certainly replace the appeal to BO with an
appeal to Lagrange multipliers (thanks for this point). • "*Correctness*": We apologize for the lack of clarity and indeed
note a random 80/20 CV split within the training dataset before application to the temporally held-out test dataset; we
will clarify in the Appendix. • "$\ell_1$ *rather than Euclidean*": We used $\ell_1$ since it has nicer properties than, e.g., $\ell_2$ in
high-dimensional spaces (Aggarwal et al., *ICDT*, 2001) but other distance metrics could be empirically quantified. •
"*Clarity*": Thanks for your detailed points; we will incorporate.

**Reviewer 2:** Thanks for recognizing our impact and your correctness points are really helpful. We will definitely
discuss embedding closeness of antonyms in the NLP setting and add an explicit remark on how LM probabilities, our
definition of (soft) grammaticality, can also encode NLP pragmatics. More generally, we will clarify our definitions of
these terms, borrowed from NLP, in the viral protein setting. We will incorporate all your detailed specific suggestions.

**Reviewer 3:** Thanks for noting the strength of our unsupervised setup and broader impacts. • "*conditionally indepen-*
*dent*": We apologize for a lack of explanation; conditional independence is by construction of the model architecture,
since we use the entire hidden-layer output as the latent variable embedding. We noted CI just to show that conditioning
on $\hat{\mathbf{z}}_i$ (or plugging in the embedding values before the final softmax layer) is sufficient to compute the final mutation
probability, so $\hat{p}(x_i|\mathbf{x}_{[N]\setminus\{i\}}, \hat{\mathbf{z}}_i) = \hat{p}(x_i|\hat{\mathbf{z}}_i)$ for the learned distribution $\hat{p}$. • "*continuous measure*": In the intervening
time, we performed additional analysis correlating the CSCS objective with continuous differential selection scores,
with CSCS also performing the best (which we can include). We do note selection scores are consistently and clearly
bimodal, indicating more or less binary escape. • "*practical implications*": Thanks for this, which is essential to discuss;
typically, a physical constraint (e.g., mutational library size or sequencing cost) directly provides a top N to acquire,
which is the best way (and already a practical one) to use CSCS predictions now. Useful further work could identify an
absolute threshold beyond which to acquire mutations (perhaps by quantifying prediction uncertainty).

**Reviewer 4:** Thanks for noting your conceptual interest and the strength of our empirical results. • "*theoretical*
*detail*"/"*how the method works*": We apologize for sparsity of detail. The fundamental reasons typically given for
how/why LMs work use an appeal to the distributional hypothesis (our paper's refs [22, 25]). Our work builds off of
recent extensions of LMs to protein sequences and is motivated by the broader impact of this approach for studying
infectious viruses. • "*how the semantic change was computed*": Sorry for the confusion here and you're right; for a
given protein sequence, we evaluate the BiLSTM at each position to obtain a sequence-length-by-embedding-dimension
matrix, which is also what previous protein embedding approaches produce (refs [9, 44, 4]). This matrix can be flattened
(as we do) or averaged across the sequence dimension to compare proteins/compute distances.• "*exhaustive search*":
Since we focus on single-token mutations, an exhaustive search (scales with alphabet-size-by-sequence-length) suffices
for our purposes, especially since current experimental validation data is also limited to single-residue mutations.
We note for (viral) proteins, alphabet size (i.e., number of natural amino acids) is constant and sequence length is
typically in the high $10^2$ or low $10^3$ range. Combinatorial search will require a different strategy, which suggests highly
interesting future work. • "*choice of beta*": We find good robustness of $\beta$ values reasonably close to 1 (e.g, 0.5-2).
Results do start to change after more than a 2X increase or decrease (e.g., when $\beta = 0.25$).

[Meta-Review · NeurIPS 2020]

This paper proposes a method for addressing the question of viral escape prediction, using an NLP-based approach. The manuscripts has been actively discussed by the referees and we agree with ultimate verdict that in particular the theoretical and conceptual contribution have considerable merit. The formulation whereby notions of semantics and grammatically are borrowed from NLP has the potential to stimulate further developments in this direction.